# Optimization of Extraction Methods for NMR and LC-MS Metabolite Fingerprint Profiling of Botanical Ingredients in Food and Natural Health Products (NHPs)

**DOI:** 10.3390/molecules30163379

**Published:** 2025-08-14

**Authors:** Varathan Vinayagam, Arunachalam Thirugnanasambandam, Subramanyam Ragupathy, Ragupathy Sneha, Steven G. Newmaster

**Affiliations:** 1College of Biological Science, University of Guelph, Guelph, ON N1G 2W1, Canada; vinaychemist89@gmail.com (V.V.); ragu@uoguelph.ca (S.R.); snehara@auamed.net (R.S.); snewmast@uoguelph.ca (S.G.N.); 2Biological and Life Sciences, Canadian Light Source, Saskatoon, SK S7N 2V3, Canada; 3College of Medicine, American University of Antigua, Jobberwock Beach Road, Coolidge P.O. Box W1451, Antigua and Barbuda

**Keywords:** nuclear magnetic resonance, liquid chromatography–mass spectrometry, metabolite fingerprinting, botanical authentication, solvent extraction, chemotaxonomy

## Abstract

Metabolite fingerprint profiling is a robust tool for verifying suppliers of authentic botanical ingredients. While comparative studies exist, few apply identical conditions across multiple species; this study utilized a cross-species comparison to identify versatile solvents despite biochemical variability. Multiple solvents were used for sample extraction prior to analysis by proton NMR and liquid chromatography–mass spectrometry (LC-MS) for multiple botanicals including *Camellia sinensis*, *Cannabis sativa*, *Myrciaria dubia*, *Sambucus nigra*, *Zingiber officinale*, *Curcuma longa*, *Silybum marianum*, *Vaccinium macrocarpon*, and *Prunus cerasus*. Comparisons were normalized by total intensity; deuterated methanol aids NMR lock but is not required for LC-MS. Hierarchical clustering analysis (HCA) evaluated solvent efficacy. Methanol–deuterium oxide (1:1) was the most effective extraction method, yielding 155 NMR spectral metabolite variables for *Camellia sinensis*, while methanol (90% CH_3_OH + 10% CD_3_OD) produced 198 for *Cannabis sativa* and 167 for *Myrciaria dubia*, with 11, 9, and 28 assigned metabolites, respectively. LC-MS detected 121 metabolites in *Myrciaria dubia* in methanol as the most effective extraction method. Methanol (10% deuterated) is the most effective extraction method for comprehensive metabolite fingerprinting using NMR and LC-MS protocols because it provides the broadest metabolite coverage. This study advances fit-for-purpose methods to qualify suppliers of botanical ingredients in food and NHP quality control programs.

## 1. Introduction

### 1.1. Why NMR for Botanical Ingredient Quality Control in NHPs and Food?

Nuclear magnetic resonance (NMR) spectroscopy is a robust and reproducible technique for ensuring the quality control of botanical ingredients utilized in food products and natural health products (NHPs). Historically, the application of NMR necessitated considerable expertise in chemistry, physics, and mathematics, restricting its use to highly specialized researchers [1,2]. Over the past decade, significant technological advancements have transformed NMR into a rapid, non-destructive, highly reproducible method, accessible to molecular laboratory technicians with minimal sample preparation requirements [3,4]. Employing a 400 MHz Bruker Avance III spectrometer, NMR facilitates the verification of botanical species across a diverse array of matrices, including medicinal plants [5] and herbs [6,7] (*Cannabis sativa* and *Zingiber officinale*); food matrices (spices [8,9], fruit juices [10,11]); and fermented products (wine [12], and beer [13]). This technique may reflect metabolomic differences supporting taxonomic identification [14], complemented by genetic methods (DNA barcoding), and determines geographic origins with high accuracy [15,16,17]. Furthermore, NMR serves as an effective tool for screening adulterants—such as fillers, added sugars, and synthetic compounds [9,18,19]—and for differentiating plant parts that are associated with specific therapeutic or nutritional efficacy claims [20,21]. Beyond identification, NMR enables the relative/absolute quantification of bioactive molecules, positioning it as a comprehensive instrument for authentication and quality assurance within the food and NHP sectors [7,22]. Recent studies have demonstrated NMR’s utility in profiling complex botanical mixtures [23,24,25,26,27], underscoring its growing relevance to industrial applications. Despite these advances, there is a need to standardize extraction methodologies to support the routine implementation of NMR quality control under good manufacturing practices (GMP) that qualify suppliers of authentic botanical ingredients. Practical challenges include cost and standardization, addressed here via optimized protocols.

### 1.2. Comparison of NMR with DNA-Based Other Analytical Chemistry Methods

NMR fingerprinting complements a spectrum of alternative authentication technologies, each with distinct strengths and limitations. DNA-based methodologies rely on species-specific genetic markers to confirm identity and detect adulteration, proving effective for minimally processed ingredients where DNA remains intact [28,29,30,31]. However, these methods encounter significant challenges with extracted materials due to the degradation of genetic material during processing [29,32]. In contrast, orthogonal analytical chemistry approaches target phytochemicals directly linked to health benefits, with targeted techniques focusing on individual compounds and non-targeted NMR offering a holistic profile of sample composition [33,34,35]. When compared to liquid chromatography–mass spectrometry (LC-MS), NMR exhibits lower sensitivity but superior reproducibility and stability [22,36]. Similar challenges affect NMR chemical shift libraries for complex mixtures. NMR, by contrast, provides a less targeted perspective on sample composition, enabling the simultaneous elucidation and relative quantification of multiple compounds regardless of the industrial processing method [7]. This enables the detection of unknown adulterants in sourced ingredients. This characteristic, coupled with its adaptability to various stages in the supply chain, has driven its increasing adoption in ensuring the quality and authenticity of food and NHPs. NMR provides a fit-for-purpose quality assurance method for non-targeted fingerprinting (authentication), targeted assignment (identification), and integration (relative quantification).

### 1.3. Key Uncertainties Concerning Different NMR Extraction Methods

The effective application of commercial NMR methods for fingerprinting botanical ingredients hinges on the optimization of extraction solvents, which selectively target specific classes of metabolites and thereby influence the accuracy of botanical species authentication [20]. For certain botanical taxa, extraction protocols incorporate solvents such as methanol, methanol blended with deuterium oxide, chloroform, and dimethyl sulfoxide, while others utilize acetone, acetonitrile, and water-based systems. Initial experimental trials employing higher sample masses demonstrated no measurable differences in extraction efficacy compared to reduced masses; consequently, the lower mass was adopted due to constraints in sample availability. Several factors complicate the extraction process, including the presence of co-extracted compounds, potential contamination, metabolite decomposition, and the formation of artifacts attributable to impurities within the solvents [2]. Variations in chemical shifts arising from pH differences can be addressed through the use of phosphate buffers in deuterium oxide, enhancing spectral consistency [37]. Certain botanicals necessitate tailored extraction strategies to mitigate issues such as enzyme activity or lipid contamination [38]. Research indicates that aqueous methanol, when supplemented with phosphate buffers, proves effective for extracting amino acids, carbohydrates, and phenolic compounds [6,39,40], while pure methanol emerges as a preferred solvent for food ingredients like olives and ginseng, offering scalability suitable for industrial protocols [20,23]. Nevertheless, a significant gap remains in the comparative analysis of extraction methods across a broad range of botanical species, underscoring the need for standardized approaches to develop comprehensive NMR spectral libraries, a challenge pertinent to the advancement of quality assurance of suppliers of authentic botanical ingredients [41].

### 1.4. Objective of This Study

This study seeks to develop standardized NMR and LC-MS extraction methodologies to enhance the authentication of botanical ingredients within the food and NHP industries. The investigation encompasses three primary model taxa alongside six additional botanical species, selected to represent a diverse range of chemical profiles and industrial relevance. The specific objectives of this research are to:Establish NMR protocols utilizing a variety of solvents to detect a broad spectrum of spectral variables across the selected taxa, adjusting sample masses based on initial trials that indicated no significant differences with varying quantities.Implement targeted LC-MS metabolomics through direct injection and reverse-phase LC-MS/MS techniques, aiming to identify and quantify an extensive array of metabolites within plant material extracts prepared at standardized concentrations.Evaluate the efficiency of different solvents in assigning metabolites through NMR-based chemical shift analyses, providing insights into extraction optimization.Conduct analyses of NMR fingerprints using hierarchical clustering techniques to group a specific model botanical (tea—*Camellia sinensis*) samples according to their key metabolite profiles, facilitating a comparative assessment of extraction solvent performance.Investigate the utility of methanol as an extraction solvent across multiple botanicals, targeting the detection of a wide range of spectral variables to assess its versatility.Examine the stability of methanol extracts over an extended duration, assessing variations in key metabolite profiles to ensure reliability under storage conditions.

## 2. Results

### 2.1. Extraction Efficiency and Sample Preparation

The extraction efficiency of nine botanical taxa was evaluated using proton NMR and LC-MS (camu camu only) protocols with comprehensive sample preparation details documented in Table 1. For *Camellia sinensis*, three distinct sample types were processed: Orange Pekoe tea extracts with a mass of 50 mg (±1 mg) using 1 mL of solvent, Green Tea extracts at 51 mg (±1 mg) with 1 mL of solvent, and Black Tea extracts at 51 mg (±1 mg) with 1 mL of solvent. For *Cannabis sativa*, dry bud samples were extracted at 50 mg (±1 mg) with 1 mL of solvent, while dry leaf samples were prepared at 51 mg (±1 mg) with 1 mL of solvent. For *Myrciaria dubia*, camu camu powder extract and dry seed samples were both processed at 300 mg (±1 mg) with 2 mL of solvent, reflecting the higher mass requirement to support LC-MS analysis, which was exclusively performed for this taxon to capture a broader metabolite profile. Additional taxa, including *Sambucus nigra* (elderberry dry fruit), *Zingiber officinale* (ginger dry root), *Curcuma longa* (turmeric dry root), *Silybum marianum* (milk thistle dry seed), *Vaccinium macrocarpon* (cranberry dry fruit), and *Prunus cerasus* (tart cherry dry fruit), were extracted at 300 mg (±1 mg) with 2 mL of solvent. The preparation process involved homogenizing the plant material to ensure uniformity, followed by solvent extraction optimized for NMR and LC-MS compatibility. The consistency in sample masses and solvent volumes across replicates facilitated reproducible spectral outputs, with NMR spectra recorded using a 0.01 ppm bin size to enhance resolution and accuracy in metabolite detection [22].

### 2.2. Metabolite Detection

Metabolite detection was systematically performed across the nine botanical taxa using proton NMR and LC-MS (camu camu only) analyses, with detailed results presented in Table 2. Spectral variables refer to binned NMR data (0.01 ppm buckets) representing non-zero intensity regions after excluding noise, solvent peaks, and zero-sum columns, followed by scaling and normalization. These variables capture meaningful chemical profiles for multivariate analysis in authentication but are not equivalent to structurally confirmed metabolites. For *Myrciaria dubia*, NMR analysis detected 167 spectral variables in methanol (90% CH_3_OH + 10% CD_3_OD), 159 in deuterium oxide (D_2_O), 165 in chloroform (CDCl_3_), 69 in acetonitrile (acetonitrile-d_3_), and 74 in acetone (acetone-d_6_). Complementary LC-MS analysis for *Myrciaria dubia* identified 121 metabolites in methanol, 80 in water, 84 in methanol–water (1:1), 53 in acetone, 41 in acetonitrile, and 48 in chloroform, providing a comprehensive profile of targeted metabolites including amino acids, organic acids, and flavonoids. For *Cannabis sativa*, NMR revealed 198 variables in methanol, 343 in deuterium oxide, 171 in chloroform, 157 in chloroform–cyclohexane (1:1), 147 in acetone, 164 in acetonitrile, 178 in dimethyl sulfoxide (DMSO-d_6_), and 181 in dimethyl sulfoxide–chloroform (2:3), reflecting a wide range of polar and nonpolar metabolites such as cannabinoids and sugars. For *Camellia sinensis*, NMR identified 82 variables in methanol, 130 in deuterium oxide, 155 in methanol–deuterium oxide (1:1), 42 in chloroform, 75 in dimethyl sulfoxide–chloroform (1:1), 126 in acetone–deuterium oxide (1:1), and 58 in acetonitrile, highlighting variations in catechin and amino acid content across tea types. Additional taxa extracted with methanol yielded *Sambucus nigra* with 397 variables, *Zingiber officinale* with 313, *Curcuma longa* with 314, *Silybum marianum* with 396, *Vaccinium macrocarpon* with 347, and *Prunus cerasus* with 402, indicating diverse metabolic profiles suitable for authentication. Metabolite assignments in NMR spectra, based on chemical shift and coupling pattern comparisons with the literature standards, are detailed in Table 3 and expanded in Table 4. For *Cannabis sativa*, 18 metabolites were assigned in methanol, 8 in water, 12 in dimethyl sulfoxide–chloroform, 10 in chloroform, 6 in acetone, and 6 in acetonitrile. For *Camellia sinensis*, 11 metabolites were assigned in methanol–water (1:1), 10 in water, 9 in acetone–deuterium oxide (1:1), 5 in chloroform, and 7 in acetonitrile. For *Myrciaria dubia*, 28 metabolites were assigned in methanol, 14 in water, 7 in acetone, and 9 in chloroform. These assignments, validated against established standards, provide specific markers for authenticating botanical ingredients and enable precise quantification for subsequent statistical analysis [7].

### 2.3. Hierarchical Clustering Analysis of Tea Samples

The distribution of featured spectral variables in *Camellia sinensis* tea samples (Orange Pekoe, Green Tea, Black Tea) was analyzed using heatmaps to evaluate differences for authentication purposes, with the results presented in Figure 1 and Figure 2. Methanol-extracted samples, with 82 variables, exhibited distinct clustering patterns that reflected variations in flavonoid and catechin content across the three tea types, providing a robust basis for species-specific identification. Water-extracted samples, with 130 variables, displayed variations in polar metabolites such as amino acids and sugars, offering valuable data for relative quantification of these compounds and distinguishing tea subtypes. The comparative heatmap in Figure 3, illustrating spectral variables ranging from 42 for *Camellia sinensis* in chloroform to 402 for *Prunus cerasus* in methanol across all botanicals, underscored solvent-dependent differences in metabolic profiles, suggesting potential markers for advanced statistical differentiation [42].

### 2.4. Stability Assessment of Methanol Extracts

The stability of methanol-extracted *Cannabis sativa* samples (dry bud and dry leaf) was thoroughly assessed over a 95-day period, with results depicted in Figure 4. Analysis of the 198 spectral variables, encompassing key cannabinoids and flavonoids, revealed a variation of less than 5%, as determined through repeated NMR measurements conducted at 0, 30, 60, and 95 days under controlled environmental conditions. This minimal variation indicated negligible degradation of active ingredients, with spectral integrity maintained across the assessment period. The stability of these extracts, particularly the consistency of metabolite signals, supports their reliability for long-term storage and authentication applications [2].

### 2.5. Heatmap Analysis of NMR Metabolite Fingerprints

Hierarchical clustering analysis (HCA) with Euclidean distance and Ward2 linkage was applied to methanol (90% CH_3_OH + 10% CD_3_OD) NMR data for nine botanicals (Table 1), visualized in Figure 3. Spectral variables ranged from 402 (*Prunus cerasus*, 100%) to 82 (*Camellia sinensis*, 20%). *Prunus cerasus* and *Sambucus nigra* (397, 99%) were clustered due to anthocyanin signals (6.5–8.0 ppm), while *Curcuma longa* (314, 78%) and *Zingiber officinale* (313, 78%) were grouped by terpenoids (1.5–2.5 ppm). *Camellia sinensis* showed weaker results with fewer variables.

### 2.6. Stability of Methanol Extracts

NMR analysis of *Cannabis sativa* methanol (90% CH_3_OH + 10% CD_3_OD) extracts over 95 days (Figure 3) showed stable chemical shifts and peak intensities. Metabolites (e.g., Δ9-tetrahydrocannabinol, 6.0–6.2 ppm; trigonelline, 8.8 ppm) exhibited intensity variations < 5% (coefficient of variation 3.2%). Signal-to-noise ratios remained at 50:1, with no peak shifts or signal loss.

## 3. Discussion

### 3.1. Chemotaxonomy and the Role of Metabolite Fingerprints in Botanical Authentication

Chemotaxonomy, the classification of plants based on their chemical composition, serves as a critical tool for species identification and the authentication of botanical ingredients [14,43]. The analysis of primary metabolites, such as sugars and amino acids, and secondary metabolites, including flavonoids and cannabinoids, enables differentiation between closely related species and tracks biochemical variations influenced by environmental or genetic factors. The results from this study, detailed in Table 2 and Table 3, underscore the utility of proton NMR fingerprinting in chemotaxonomic applications, generating reproducible spectral data that reveal species-specific metabolite patterns [9]. For instance, the detection of 198 spectral variables in *Cannabis sativa* methanol extracts and 82 in *Camellia sinensis* methanol extracts highlights distinct metabolic profiles that can be used to authenticate these taxa. Proton NMR spectroscopy, with its high reproducibility and minimal sample preparation that is non-destructive (i.e., samples can be stored and re-analyzed), has proven to be a valuable method for verifying the identity of medicinal plants (e.g., *Cannabis sativa* via assigned metabolites) [14,44], herbs and spices (e.g., *Zingiber officinale* via 313 variables) [6,8], and food commodities (e.g., *Camellia sinensis* via 82 variables) [11,15]. Furthermore, NMR facilitates the differentiation of plant parts (e.g., *Cannabis sativa* dry bud vs. dry leaf) and the detection of adulterants, enhancing its role in quality control [18,24]. The specific metabolite assignments, such as 18 for *Cannabis sativa* in methanol and 28 for *Myrciaria dubia* in methanol (Table 3), provide chemotaxonomic markers that can be utilized for statistical analysis to authenticate botanical ingredients. However, the selection of extraction solvents significantly influences the detection and quantification of these metabolites. Therefore, extraction protocols need to maximize the detection and sensitivity of these metabolites for authentication. The rationale for this study focused on standardizing protocols for diverse taxa. This study focuses on multiple species; we suggest that future work should focus on intraspecific commercial varieties sourced as ingredients.

### 3.2. Comparison of Extraction Methods for Cannabis Samples

The comparison of NMR spectra for *Cannabis sativa* extracts, as presented in Table 2, revealed significant solvent-dependent variations. Water extraction yielded the highest number of spectral variables (343; 0.01 ppm buckets, excluding noise/solvent), indicating its efficacy in capturing polar metabolites such as sugars and amino acids, followed by methanol (198 variables), with a broader profile including cannabinoids and flavonoids. Dimethyl sulfoxide (178 variables) and dimethyl sulfoxide–chloroform (181 variables) enhanced the detection of nonpolar cannabinoids, while chloroform (171 variables) and acetone (147 variables) showed moderate efficiency. The high variable count in water and methanol extracts suggests their superior ability to capture diverse metabolites, with water excelling in polar compounds and methanol providing comprehensive coverage of both polar and nonpolar metabolites, including key authentication markers such as those assigned in methanol (Table 3). Chloroform-based solvents, despite detecting 171 variables, exhibited lower resolution due to baseline rolling issues from solvent volatility, limiting their utility for polar metabolite quantification [3]. These findings indicate that methanol extracts, with their balanced profile and 18 assigned metabolites, are the most effective for authenticating *Cannabis sativa* and relative quantification of its bioactive compounds for statistical modeling.

### 3.3. Comparison of Extraction Methods for Camu Camu Samples

NMR spectra of camu camu (*Myrciaria dubia*) revealed that methanol extraction provided the highest variable count (167; distinct bins post-processing), encompassing a wide range of metabolites, followed by chloroform (165) and water (159), with acetonitrile (69) and acetone (74) yielding fewer variables (Table 2). The LC-MS analysis further revealed 121 metabolites in methanol, 80 in water, and 84 in methanol–water (1:1), indicating methanol’s superior efficiency for capturing diverse metabolites, including those assigned as 28 in methanol and 14 in water (Table 3). Methanol was effective in extracting both polar (e.g., organic acids) and nonpolar (e.g., flavonoids) compounds, while water was effective in extracting polar metabolites. Chloroform had a lower efficiency (165 variables) and baseline issues that restrict its use for comprehensive profiling of botanicals [3]. Methanol extraction captured many metabolites, which was supported by the LC-MS data. This suggests that methanol should be considered as the preferred solvent for *Myrciaria dubia* ingredient authentication, with water serving as a complementary option for polar metabolite analysis.

### 3.4. Comparison of Extraction Methods for Tea Samples

NMR spectra of *Camellia sinensis* tea extracts indicated that methanol–deuterium oxide (1:1) and water extracts yielded the highest number of metabolite variables (155 and 130, respectively). The other extraction methods extracted fewer metabolite variables: acetone–deuterium oxide (1:1) yielded 126 variables; dimethyl sulfoxide–chloroform (1:1) yielded 75 variables; and chloroform yielded 42 variables (Table 2). The heatmaps in Figure 1 and Figure 2 revealed distinct clusters, with methanol–deuterium oxide and water extracts showing diverse metabolite profiles, including those assigned as 11 in methanol–water (1:1) and 10 in water (Table 3). Chloroform-based extracts were grouped separately due to lower diversity in the metabolite profiles [42]. This suggests that methanol–deuterium oxide is the most effective solvent for comprehensive *Camellia sinensis* fingerprinting, with water providing a focused profile for polar metabolites, facilitating authentication across all types of tea (Orange Pekoe, Green Tea, Black Tea).

### 3.5. Heatmap Analysis of Metabolite Fingerprints

The heatmaps (Figure 1, Figure 2 and Figure 3) illustrate the distribution of spectral similarities in metabolite composition, providing a visual basis for ingredient authentication. Water extracts, with 130 variables for *Camellia sinensis*, showed high sugar and amino acid signals, indicating limited discriminatory power due to homogeneity across tea types, reducing their utility for chemotaxonomic differentiation of botanical species. In contrast, methanol extracts, with 82 variables for *Camellia sinensis* and up to 402 for *Prunus cerasus*, captured a diverse set of polar and semi-polar compounds, facilitating species-specific identification. The interspecific variation was considerable, allowing grouping by species (e.g., *Cannabis sativa* via 198 variables, *Myrciaria dubia* via 167 variables), while intraspecific variation remained low, supporting the reliability of methanol extracts for authentication. The broad range of variables (42–402) in Figure 3 underscores the solvent’s impact on metabolic diversity, offering a rich dataset for multivariate statistical analysis to authenticate botanical ingredients [42].

### 3.6. Effect of Extraction Solvent on Metabolite Fingerprint Profiles

The extraction methods significantly influenced metabolite detection and relative quantification, as evidenced by the variable counts in Table 2. Methanol–deuterium oxide captured a comprehensive profile, including flavonoids and catechins in *Camellia sinensis* (155 variables), while deuterium oxide excelled in polar metabolites such as sugars and amino acids (130 variables). The LC-MS analysis of *Myrciaria dubia* quantified 121 metabolites in methanol, with concentrations varying by solvent, reinforcing the need for standardized protocols [22]. The variation in metabolites captured emphasizes the importance of solvent selection in optimizing methods for acquiring metabolite fingerprints that can be statistically analyzed. This provides a standard, fit-for-purpose quality assurance method that can be effectively used to verify the authenticity of botanical ingredients.

### 3.7. Comparison of Solvent Efficiency for Metabolite Extraction

Methanol and methanol–water (1:1) extracts yielded the highest metabolite counts, with 167 NMR variables. Methanol extraction was also high using LC-MS, capturing 121 metabolites for *Myrciaria dubia* in methanol with a preference for amino acids, organic acids, and flavonoids (around 60% overlap with water for polar classes). Water extracts captured 159 NMR variables and 80 metabolites using LC-MS with a preference for sugars and hydrophilic compounds. Acetone and acetonitrile extracts were less effective, capturing 74 and 69 NMR variables, respectively, targeting lipid-soluble compounds. Chloroform captured 165 NMR variables with a preference for lipophilic metabolites [22]. Overall, methanol was effective in capturing many metabolites from the most diverse broad range of chemical groups, providing a comprehensive metabolite fingerprint profile for the model taxa evaluated in this study.

### 3.8. Efficiency of Methanol Extraction on Other Botanicals

Methanol extraction efficiency was assessed for six additional botanicals, with the results shown in Table 2. *Sambucus nigra* yielded 397 variables, reflecting a rich profile of anthocyanins and phenolic acids. *Zingiber officinale* produced 313 variables, capturing anti-inflammatory gingerols and related compounds. *Curcuma longa* yielded 314 variables, including curcuminoids with therapeutic potential. *Silybum marianum* yielded 396 variables, extracting silymarin and flavonolignans. *Vaccinium macrocarpon* yielded 347 variables, encompassing proanthocyanidins and flavonoids. *Prunus cerasus* yielded 402 variables, capturing anthocyanins and phenolic acids. These variable counts demonstrate methanol’s consistent efficacy for the authentication and relative quantification of diverse botanical matrices for all the taxa in this study [20].

### 3.9. Metabolite Peak Assignments

Metabolite peak assignments varied by solvent, providing specific markers for authentication (Table 3 and Table 4). For *Cannabis sativa*, 18 metabolites were assigned in methanol, 8 in water, 12 in dimethyl sulfoxide–chloroform, 10 in chloroform, 6 in acetone, and 6 in acetonitrile, including cannabinoids and sugars. For *Camellia sinensis*, 22 metabolites were assigned in methanol, 11 in methanol–water (1:1), 10 in water, 9 in acetone–deuterium oxide (1:1), 5 in chloroform, and 6 in acetonitrile, including catechins and amino acids. For *Myrciaria dubia*, 28 metabolites were assigned in methanol, 14 in water, 7 in acetone, and 9 in chloroform, encompassing organic acids and flavonoids. These are representative assignments based on chemical shift analysis, offering botanical-specific markers for chemotaxonomic authentication (reliability is assessed via comparison to synthetic compounds, HMDB, and the literature, with biological relevance) and, in some cases, the verification of metabolite claims on botanical product labels [7]. The peak assignments of the NMR spectral images are shown in Appendix A.

### 3.10. Extract Stability During Storage and Ingredient Stability over Time

The stability of methanol-extracted *Cannabis sativa* samples over 95 days is shown in Figure 4. Degradation was low, with less than 5% variation among the extracted 198 spectral variables. We repeated NMR measurements at 0, 30, 60, and 95 days under controlled conditions. The results indicate that the metabolite signals were stable, suggesting resistance to hydrolysis and oxidation (qualitative; monitored key peaks THC and CBD). This was important for *Cannabis sativa* samples, where cannabinoids were stable through time, which is useful for product label claims concerning specific cannabinoids. The stability of methanol extraction outperforms volatile solvents like chloroform, which exhibited baseline issues [2]. The reliability of long-term methanol extract preservation is important for quality control. The consistent integrity of assigned metabolites (e.g., nine in methanol) ensures reproducibility in authentication protocols. No major changes indicate that the polar metabolites are stable.

### 3.11. Overview and Implications

The results demonstrate that methanol-based solvents, particularly methanol–deuterium oxide, provide the most comprehensive metabolite profiles, with specific assignments (e.g., 9 for *Cannabis sativa*, 28 for *Myrciaria dubia*) serving as chemotaxonomic markers for ingredient authentication. The stability of methanol extracts over 95 days, with <5% variation, supports their adoption in GMP-compliant quality control, addressing industry needs for reproducible botanical authentication [20]. The integration of NMR and LC-MS data offers a robust framework for developing standardized extraction protocols, enhancing authentication and relative quantification in food and natural health product industries.

## 4. Materials and Methods

### 4.1. Experimental Design and Sample Collection

The study was designed to evaluate the extraction efficiency and metabolite profiles for nine botanical taxa using proton nuclear magnetic resonance (NMR) spectroscopy and liquid chromatography–mass spectrometry (LC-MS). The evaluation was focused on the need for extraction methods that can provide diverse metabolomic fingerprint profiles that would be useful for chemotaxonomic authentication of botanical species ingredients. The selected taxa included three model species—*Camellia sinensis* (tea varieties: Orange Pekoe, Green Tea, Black Tea), *Cannabis sativa* (dry bud, dry leaf), and *Myrciaria dubia* (camu camu powder extract, dry seed)—along with six additional species: *Sambucus nigra* (elderberry dry fruit), *Zingiber officinale* (ginger dry root), *Curcuma longa* (turmeric dry root), *Silybum marianum* (milk thistle dry seed), *Vaccinium macrocarpon* (cranberry dry fruit), and *Prunus cerasus* (tart cherry dry fruit). Dried plant material was obtained from certified commercial suppliers and authenticated by botanical experts using morphological and genetic markers prior to analysis. Samples were stored at 4 °C in airtight, light-protected containers to prevent degradation until processing. The experimental design employed multiple deuterated and non-deuterated solvent systems to optimize metabolite extraction, with LC-MS analysis restricted to *Myrciaria dubia* to broaden metabolite coverage [22]. All sample vouchers are stored at the NHP Research Alliance, at the University of Guelph, Ontario, Canada. Cannabis research was conducted following University of Guelph research license (LIC-ANQAC4M3XI-2020) including handling and destruction of all samples according to the license requirements. LC-MS was limited to *Myrciaria dubia* for in-depth profiling and consistent solvent testing across taxa for NMR. All conditions were tested across model taxa and subsets for others due to relevance.

### 4.2. Sample Preparation

Sample preparation protocols were optimized for uniformity and reproducibility. Dried plant material (2 g) was homogenized using an IKA (Wilmington, NC, USA) Tube Mill (40 mL chamber, 25,000 rpm) for 2 min to achieve a consistent particle size. *Camellia sinensis*, Orange Pekoe tea, Green Tea, and Black Tea extracts were prepared at 50 mg (±1 mg) and 51 mg (±1 mg), respectively, using 1.0 mL of solvent. *Cannabis sativa* dry bud and dry leaf samples were extracted at 50 mg (±1 mg) and 51 mg (±1 mg) with 1.0 mL of solvent. The remaining taxa, including *Myrciaria dubia* (camu camu) and six additional species powder extracts and dry seeds, were extracted at 300 mg (±1 mg) of material with 2.0 mL of solvent to support LC-MS analysis where applicable. All extractions were performed in triplicate to account for any sample variation. Each sample was vortexed for 30 s, sonicated for 15 min at room temperature (25 °C), and centrifuged at 13,000× *g* for 10 min to collect the supernatant. Quality control (QC) samples were prepared by pooling equal volumes of each taxon’s extracts to monitor batch consistency, while blank samples (solvent only) were analyzed to assess background signals. Daily calibration of the NMR spectrometer with an external standard (0.1% trimethylsilane in methanol) ensured instrumental accuracy.

### 4.3. Selection of Extraction Solvents

Solvent selection was tailored to extract chemotaxonomically relevant metabolites, ensuring well-resolved NMR and LC-MS peaks for untargeted and targeted chemometric analysis [4,20]. Deuterated solvents were used for NMR to provide a lock signal, with methanol comprising 90% CH_3_OH + 10% CD_3_OD, while other NMR solvents (e.g., D_2_O, CDCl_3_) were fully deuterated. Non-deuterated solvents were employed for LC-MS analysis of *Myrciaria dubia*. The solvent systems targeted specific metabolite classes:Methanol (90% CH_3_OH + 10% CD_3_OD): Extracted polar and semi-polar metabolites, including alkaloids, flavonoids, and cannabinoids [43].Deuterium Oxide (D_2_O): As a dissolution medium for polar extracts such as amino acids and sugars [37].Chloroform (CDCl_3_): Used for the extraction of lipophilic compounds, including lipids and terpenoids [43].Dimethyl Sulfoxide (DMSO-d_6_): Enhanced extraction of flavonoids and alkaloids [20].Acetone (Acetone-d_6_) and Acetonitrile (Acetonitrile-d_3_): Targeted phenolics and flavonoids [45].Combinations: Included methanol–deuterium oxide (1:1), chloroform–cyclohexane (1:1), dimethyl sulfoxide–chloroform (2:3 and 1:1), and acetone–deuterium oxide (1:1) to broaden metabolite coverage.

LC-MS analysis of *Myrciaria dubia* utilized non-deuterated solvents including methanol, water, methanol–water (1:1), acetone, acetonitrile, and chloroform. The effectiveness of these solvent systems in detecting spectral variables and metabolites is summarized in Table 2.

### 4.4. Proton NMR Acquisition and Processing

Proton NMR spectra were acquired using a Bruker (Billerica, MA, USA) Avance III 400 MHz spectrometer equipped with a 5 mm broadband probe (room temperature), employing a NOESY-based 1D presaturation experiment to suppress the water signal. Acquisition parameters included a 12 ppm spectral width, 64k time domain points, a 4 s relaxation delay, and 128 scans to achieve signal-to-noise ratios ranging from 20:1 to 60:1. Spectra were processed using Bruker TopSpin 4.1 software, involving Fourier transformation with 2-fold zero-filling, the application of 0.3 Hz exponential line broadening, automated and manual phase correction, and third-order polynomial baseline correction. Chemical shifts were referenced to 0.1% trimethylsilane (TMS) in methanol at 0.00 ppm. Spectra were binned into 0.01 ppm buckets and normalized to total spectral intensity, with noise reduction achieved by zeroing values below the mean intensity and filtering zero-sum columns. Metabolite identification was performed by comparing chemical shifts and coupling patterns with the Human Metabolome Database (HMDB) and the published literature [2,46]. The number of assigned metabolites for model taxa is detailed in the relevant table. Hierarchical clustering analysis (HCA) was conducted using R (version 4.2.3) with Euclidean distance and Ward2 linkage on methanol-extracted data to differentiate taxa [20].

### 4.5. Extract Solvent Stability During Storage

To assess the stability of extracted samples, aliquots of the prepared extracts were stored in airtight amber vials to prevent photodegradation and maintained at 4 °C in a refrigerator for specified durations (0, 30, 60, and 95 days) to evaluate long-term integrity. For data collection, samples were removed from storage and allowed to equilibrate to room temperature (25 °C) for 30 min. Each sample was then placed in a sonicating bath operating at 40 Hz for 30 s to ensure homogeneity prior to analysis. Subsequently, the samples were transferred to 5 mm NMR tubes, and NOESY proton NMR data were collected using the same acquisition parameters as described above. This procedure minimized disruption to metabolite profiles during stability assessment, with spectral changes monitored to quantify degradation over time [2].

### 4.6. LC-MS Analysis

Targeted metabolomics for *Myrciaria dubia* (camu camu powder extract and dry seed) was performed using an Agilent (Santa Clara, CA, USA) 1260 UHPLC system coupled to an ABSciex (Framingham, MA, USA) 4000 QTrap mass spectrometer, equipped with a reverse-phase C18 column (2.1 mm × 100 mm, 1.8 µm particle size). Samples (300 mg) were extracted in 2.0 mL of solvent (methanol, water, methanol–water 1:1, acetone, acetonitrile, chloroform). The LC-MS/MS assay targeted up to 121 metabolites, including amino acids, organic acids, phenolic compounds, and flavonoids, with optimized conditions including a flow rate of 0.3 mL/min, column temperature of 40 °C, and a gradient elution from 5% to 95% acetonitrile with 0.1% formic acid over 20 min [46]. Non-acidic organic metabolites were derivatized with phenyl-isothiocyanate (enhances ionization for low-abundance analytes), extracted with 5 mM ammonium acetate in methanol, and diluted with 50:50 methanol–water (0.1% formic acid). Organic acids were precipitated with ice-cold methanol, derivatized with 3-nitrophenylhydrazine (targets carboxylic acids), and stabilized with 0.01% butylated hydroxytoluene (BHT; prevents oxidation, validated in pilot tests). Mass spectrometry operated in positive and negative ion modes with multiple reaction monitoring (MRM) pairs, using isotope-labeled internal standards for relative quantification (standards for amino acids/carboxylic acids; no absolute data presented). Data were processed with Analyst 1.6.2 software, comparing retention times and MRM transitions to Sigma-Aldrich standards and the Metlin database (identifications via exact mass, RT, fragmentation; criteria per Metlin). Quality control samples, injected every 10 samples, exhibited peak area relative standard deviations below 15%. Solvent efficiency was evaluated using MetaboAnalyst 5.0, with statistical analyses pending optimization [47].

### 4.7. Statistical Analysis

NMR data were preprocessed by log-transformation and Pareto-scaling in MetaboAnalyst 5.0 and R (version 4.2.3) to normalize variance. Hierarchical clustering analysis (HCA) was applied to methanol-extracted spectral variables, using Euclidean distance and Ward2 linkage, to differentiate the nine botanical taxa. LC-MS data for *Myrciaria dubia* were similarly processed in MetaboAnalyst 5.0, with statistical methods (e.g., principal component analysis, partial least squares-discriminant analysis) pending optimization to assess solvent efficiency and metabolite differentiation [20,46].

## 5. Conclusions

This study systematically evaluated extraction solvents for metabolite fingerprinting of nine botanical taxa using proton nuclear magnetic resonance (NMR) spectroscopy and liquid chromatography–mass spectrometry (LC-MS), as detailed in the experimental methods and summarized in the relevant tables. The findings emphasize the critical role of solvent selection in optimizing metabolite extraction for the authentication and quality control of food and natural health products (NHPs), addressing industry needs for standardized methods that support regulatory compliance [20].

### 5.1. Implications for Botanical Ingredient Authentication

The results highlight the importance of tailored solvent selection for comprehensive metabolite fingerprinting. Methanol (90% CH_3_OH + 10% CD_3_OD) emerged as the most effective primary NMR solvent across the nine botanicals, detecting 198 spectral variables for *Cannabis sativa*, 167 for *Myrciaria dubia*, and 82 for *Camellia sinensis*, capturing diverse metabolites such as flavonoids, alkaloids, and organic acids. Methanol–deuterium oxide (1:1) proved most effective for *Camellia sinensis*, yielding 155 variables and effectively detecting catechins and amino acids. Dimethyl sulfoxide-based solvents (e.g., 181 variables in dimethyl sulfoxide–chloroform 2:3 for *Cannabis sativa*) and chloroform (171 variables for *Cannabis sativa*) enhanced the extraction of specific classes like cannabinoids and lipophilic compounds, though chloroform’s utility was limited by baseline distortions in 10–15% of spectra [18]. For LC-MS, methanol extracted 121 metabolites from *Myrciaria dubia*, with methanol–water (1:1) yielding 84 metabolites, providing a robust profile for targeted metabolomics. The development of standardized extraction and analysis protocols, informed by these solvent-specific outcomes, is essential for establishing consistent NMR and LC-MS spectral libraries [9,15].

### 5.2. Key Findings

Solvent Efficiency: Methanol (90% CH_3_OH + 10% CD_3_OD) demonstrated high efficiency, capturing 198 variables (58% of maximum) for *Cannabis sativa*, 167 variables (100%) for *Myrciaria dubia*, and 82 variables for *Camellia sinensis*, encompassing flavonoids, alkaloids, and organic acids (based on variable counts; criteria: >150 variables, <10% CV reproducibility). Methanol–deuterium oxide (1:1) was optimal for *Camellia sinensis* with 155 variables (100%), effectively detecting catechins and alkaloids. For LC-MS, methanol extracted 121 metabolites (100%) from *Myrciaria dubia*, followed by methanol–water (1:1) with 84 metabolites (69%) [22].Polar Metabolite Extraction: Deuterium oxide (D_2_O) excelled in extracting polar metabolites, yielding 343 variables (100%) for *Cannabis sativa* and 159 variables (95%) for *Myrciaria dubia*, though its limited coverage of nonpolar compounds reduced its chemotaxonomic utility [37].Nonpolar Metabolite Extraction: Chloroform targeted lipophilic metabolites, detecting 171 variables (50%) for *Cannabis sativa* and 165 variables (99%) for *Myrciaria dubia*, but baseline distortions in 10–15% of spectra compromised reliability, necessitating alternative solvents [24,43].Chemotaxonomic Applications: Methanol-based NMR data facilitated robust chemotaxonomic libraries via hierarchical clustering analysis (HCA), differentiating the nine taxa: *Prunus cerasus* (402 variables), *Sambucus nigra* (397), *Silybum marianum* (396), *Vaccinium macrocarpon* (347), *Curcuma longa* (314), *Zingiber officinale* (313), *Cannabis sativa* (198), *Myrciaria dubia* (167), and *Camellia sinensis* (82). This approach supports species-specific authentication [9].Stability of Methanol Extraction: Methanol extracts of *Cannabis sativa*, stored in airtight amber vials at 4 °C for 95 days, maintained spectral integrity with less than 5% variation in 198 variables. The pre-analysis protocol—equilibration to room temperature (25 °C) for 30 min and sonication at 40 Hz for 30 s before NMR tube transfer—ensured minimal disruption, validating long-term quality control applicability [2,20].Complementary LC-MS Analysis: LC-MS detected 121 metabolites in *Myrciaria dubia* with methanol, though approximately 30% remain unassigned. Expansion to all nine taxa is planned, with statistical methods (e.g., partial least squares–discriminant analysis) under optimization [47].

### 5.3. Future Directions

Future research should investigate alternative solvents, such as dichloromethane, to mitigate chloroform’s baseline issues (10–15%) and improve nonpolar metabolite detection [22]. Extending LC-MS analysis to all nine taxa, once methods and metabolite assignments are optimized, will provide a holistic metabolite profile. Advanced chemometric techniques, including partial least squares–discriminant analysis (PLS-DA) and Random Forests, alongside automated NMR processing, are recommended to enhance authentication efficiency for food and NHP industries [9,24]. The assemblage of NMR metabolite fingerprints for many botanical species ingredients using these extraction methods will provide a standard protocol for quality assurance programs. This would advance fit-for-purpose methods to qualify suppliers of botanical ingredients in food and NHP quality control programs.

### 5.4. Conclusion Statement

Methanol (90% CH_3_OH + 10% CD_3_OD) and methanol–deuterium oxide (1:1) are the most effective NMR solvents for botanical fingerprinting, offering comprehensive metabolite coverage, reproducibility, and stability over 95 days with less than 5% variation. Methanol and methanol–water (1:1) excel for LC-MS profiling of *Myrciaria dubia*, with plans to extend this approach. HCA-based chemotaxonomic libraries enable reliable botanical species ingredient authentication, addressing regulatory and safety requirements for the food and NHPs [9,20].

## Figures and Tables

**Figure 1 molecules-30-03379-f001:**
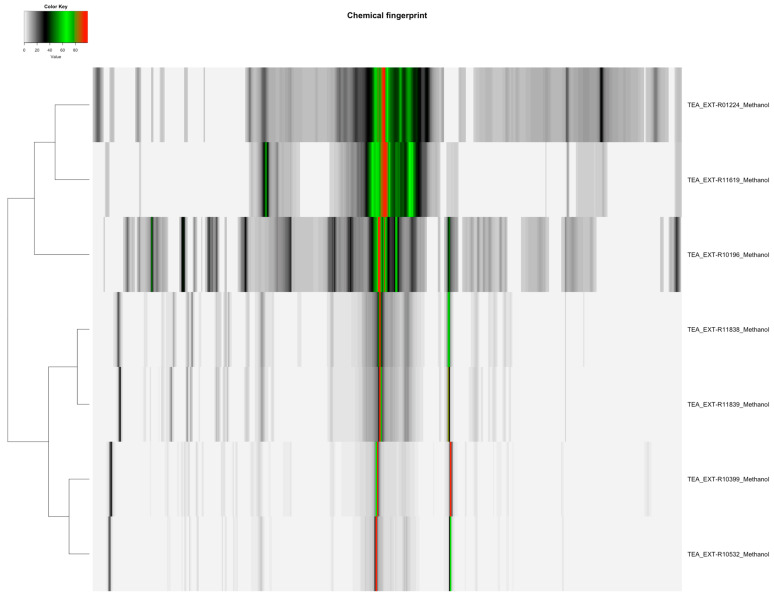
Heatmap of methanol-extracted *Camellia sinensis* tea samples (Orange Pekoe, Green Tea, Black Tea), displaying 82 spectral variables, illustrating the distribution and differences in flavonoid and catechin content across the samples for authentication purposes.

**Figure 2 molecules-30-03379-f002:**
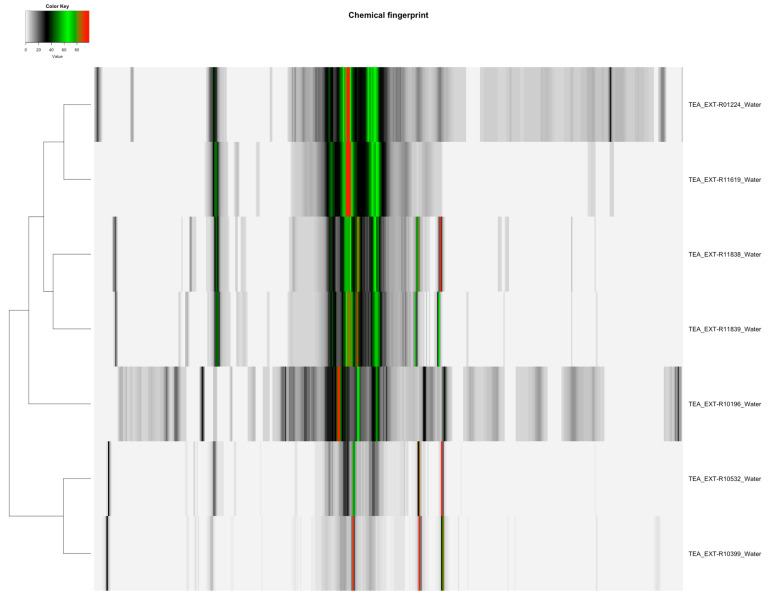
Heatmap of water-extracted *Camellia sinensis* tea samples (Orange Pekoe, Green Tea, Black Tea), displaying 130 spectral variables, illustrating the distribution and differences in polar metabolites such as amino acids and sugars across the samples for quantification.

**Figure 3 molecules-30-03379-f003:**
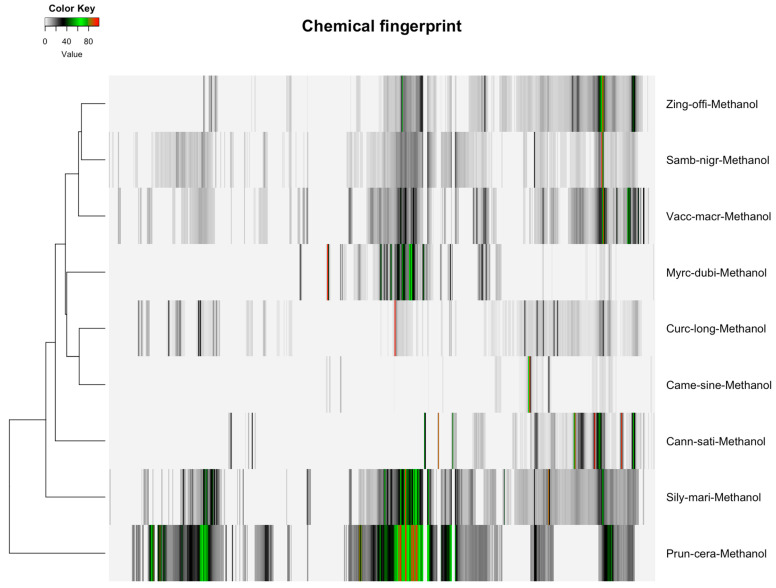
Heatmap of detected spectral variables for botanical extracts across nine taxa, ranging from 42 variables for *Camellia sinensis* in chloroform to 402 variables for *Prunus cerasus* in methanol, highlighting variability in metabolic profiles influenced by solvent selection for statistical analysis.

**Figure 4 molecules-30-03379-f004:**
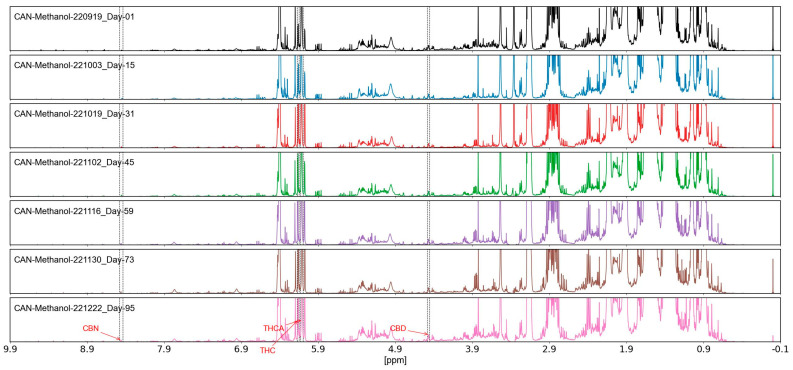
Stability assessment of methanol-extracted *Cannabis sativa* samples (dry bud and dry leaf) over 95 days, showing less than 5% variation in 198 spectral variables, minor shifts < 0.01 ppm were negligible; CV 3.2%, measured via NMR under controlled conditions.

**Table 1 molecules-30-03379-t001:** Sample details for botanical extractions, listing sample types, corresponding taxa (*Camellia sinensis*, *Cannabis sativa*, *Myrciaria dubia*, *Sambucus nigra*, *Zingiber officinale*, *Curcuma longa*, *Silybum marianum*, *Vaccinium macrocarpon*, *Prunus cerasus*), masses of homogenized plant material (±1 mg tolerance), and solvent volumes used for proton NMR and LC-MS extractions, with LC-MS analysis performed only for *Myrciaria dubia* (camu camu product and seed).

Sample Type	Taxon	Mass (±1 mg)	Solvent Volume (mL)
Tea—Orange Pekoe (fermented; extracts)	*Camellia sinensis*	50	1
Tea—Green Tea (unfermented; extracts)	*Camellia sinensis*	51	1
Tea—Black Tea (fermented; extracts)	*Camellia sinensis*	51	1
Cannabis—dry bud	*Cannabis sativa*	50	1
Cannabis—dry leaf	*Cannabis sativa*	51	1
Camu camu—powder extract	*Myrciaria dubia*	300	2
Camu camu—dry seed	*Myrciaria dubia*	300	2
Elderberry—dry fruit	*Sambucus nigra*	300	2
Ginger—dry root	*Zingiber officinale*	300	2
Turmeric—dry root	*Curcuma longa*	300	2
Milk thistle—dry seed	*Silybum marianum*	300	2
Cranberry—dry fruit	*Vaccinium macrocarpon*	300	2
Tart cherry—dry fruit	*Prunus cerasus*	300	2

**Table 2 molecules-30-03379-t002:** Detected variables and metabolites in NMR and LC-MS analyses, detailing the number of spectral variables (NMR, 0.01 ppm bin size) and metabolites (LC-MS) for *Cannabis sativa*, *Camellia sinensis*, *Myrciaria dubia*, and six additional botanicals across various deuterated solvent systems, with LC-MS data specific to *Myrciaria dubia*.

Taxon	Analysis	Solvent System	NMR Variables	LC-MS Metabolites
*Myrciaria dubia*	NMR	Methanol (90% CH_3_OH + 10% CD_3_OD)	167	N/A
*Myrciaria dubia*	NMR	Deuterium oxide (D_2_O)	159	N/A
*Myrciaria dubia*	NMR	Chloroform (CDCl_3_)	165	N/A
*Myrciaria dubia*	NMR	Acetonitrile (Acetonitrile-d_3_)	69	N/A
*Myrciaria dubia*	NMR	Acetone (Acetone-d_6_)	74	N/A
*Myrciaria dubia*	LC-MS	Methanol	N/A	121
*Myrciaria dubia*	LC-MS	Water	N/A	80
*Myrciaria dubia*	LC-MS	Methanol–water (1:1)	N/A	84
*Myrciaria dubia*	LC-MS	Acetone	N/A	53
*Myrciaria dubia*	LC-MS	Acetonitrile	N/A	41
*Myrciaria dubia*	LC-MS	Chloroform	N/A	48
*Cannabis sativa*	NMR	Methanol (90% CH_3_OH + 10% CD_3_OD)	198	N/A
*Cannabis sativa*	NMR	Deuterium oxide (D_2_O)	343	N/A
*Cannabis sativa*	NMR	Chloroform (CDCl_3_)	171	N/A
*Cannabis sativa*	NMR	Chloroform–cyclohexane (1:1)	157	N/A
*Cannabis sativa*	NMR	Acetone (Acetone-d_6_)	147	N/A
*Cannabis sativa*	NMR	Acetonitrile (Acetonitrile-d_3_)	164	N/A
*Cannabis sativa*	NMR	Dimethyl sulfoxide (DMSO-d_6_)	178	N/A
*Cannabis sativa*	NMR	Dimethyl sulfoxide–chloroform (2:3)	181	N/A
*Camellia sinensis*	NMR	Methanol (90% CH_3_OH + 10% CD_3_OD)	82	N/A
*Camellia sinensis*	NMR	Deuterium oxide (D_2_O)	130	N/A
*Camellia sinensis*	NMR	Methanol–deuterium oxide (1:1)	155	N/A
*Camellia sinensis*	NMR	Chloroform (CDCl_3_)	42	N/A
*Camellia sinensis*	NMR	Dimethyl sulfoxide–chloroform (1:1)	75	N/A
*Camellia sinensis*	NMR	Acetone–deuterium oxide (1:1)	126	N/A
*Camellia sinensis*	NMR	Acetonitrile (Acetonitrile-d_3_)	58	N/A
*Sambucus nigra*	NMR	Methanol (90% CH_3_OH + 10% CD_3_OD)	397	N/A
*Zingiber officinale*	NMR	Methanol (90% CH_3_OH + 10% CD_3_OD)	313	N/A
*Curcuma longa*	NMR	Methanol (90% CH_3_OH + 10% CD_3_OD)	314	N/A
*Silybum marianum*	NMR	Methanol (90% CH_3_OH + 10% CD_3_OD)	396	N/A
*Vaccinium macrocarpon*	NMR	Methanol (90% CH_3_OH + 10% CD_3_OD)	347	N/A
*Prunus cerasus*	NMR	Methanol (90% CH_3_OH + 10% CD_3_OD)	402	N/A

**Table 3 molecules-30-03379-t003:** Assigned metabolites in NMR spectra by solvent system, providing the number of metabolites assigned in proton NMR spectra for *Cannabis sativa* (Cannabis), *Camellia sinensis* (Tea), and *Myrciaria dubia* (camu camu) across selected solvent systems, based on chemical shift and coupling pattern comparisons with the literature standards.

Solvent System	Cannabis	Tea	Camu Camu
Methanol	18	22	28
Methanol–water (1:1)	Not tested	11	Not tested
Water	8	10	14
Acetone–deuterium oxide (1:1)	Not tested	9	Not tested
Acetone	6	Not tested	7
Dimethyl sulfoxide–chloroform	12	7	Not tested
Chloroform	10	5	9
Acetonitrile	6	6	6

**Table 4 molecules-30-03379-t004:** NMR spectral metabolite peak assignments for model taxa. Assignments are provided for key metabolites in a methanol solvent; however, not all variables are assigned.

No.	Species	Compound Class	Metabolite	^1^H NMR Chemical Shift (δ, ppm)
1	Tea	Alkaloids	Caffeine	7.74 (s, 1H), 3.89 (s, 3H), 3.54 (s, 3H), 3.29 (s, 3H)
2	Tea	Alkaloids	Theobromine	7.77 (s, 1H), 3.46 (s, 3H)
3	Tea	Phenolic compounds	p-Coumaroyl quinic acid	7.47 (s, 1H), 7.36 (s, 1H)
4	Tea	Phenolic compounds	Gallic acid	7.01 (s, 2H)
5	Tea	Phenolic compounds	Quinic acid	4.17–4.19 (m, 2H), 1.99–2.05 (m, 4H)
6	Tea	Flavonoids	Epigallocatechin-3-gallate	6.87 (s, 1H), 6.50 (s, 1H), 5.82 (d, *J* = 2.3 Hz, 1H), 5.50 (m, 2H), 2.70–2.75 (m, 4H)
7	Tea	Flavonoids	Epigallocatechin	6.75 (s, 1H), 6.50 (s, 1H), 5.86 (d, *J* = 2.3 Hz, 1H), 5.49 (m, 1H), 2.81–2.86 (m, 2H)
8	Tea	Flavonoids	Epicatechin-3-gallate	6.90 (s, 1H), 5.90 (d, *J* = 2.3 Hz, 1H), 5.51 (m, 2H), 2.91–2.94 (m, 2H)
9	Tea	Flavonoids	Epicatechin	5.93 (s, 1H), 2.96 (m, 2H)
10	Tea	Flavonoids	Catechin	6.86 (s, 1H), 5.93 (s, 2H), 2.99 (m, 2H)
11	Tea	Carbohydrates	Sucrose	5.39 (d, *J* = 3.8 Hz, 1H)
12	Tea	Carbohydrates	α-Glucose	5.16 (d, *J* = 3.5 Hz, 1H)
13	Tea	Carbohydrates	β-Glucose	4.50 (d, *J* = 7.3 Hz, 1H)
14	Tea	Carbohydrates	Fructose	4.12 (d, *J* = 8.2 Hz, 1H)
15	Tea	Amino acids	Threonine	3.50 (d, *J* = 4.0 Hz, 1H)
16	Tea	Amino acids	Valine	0.93 (m, 3H)
17	Tea	Amino acids	Isoleucine	0.98 (m, 3H)
18	Tea	Amino acids	Leucine	1.12 (m, 3H)
19	Tea	Other compounds	Succinic acid	2.15 (s, 4H)
20	Tea	Other compounds	Acetic acid	1.99 (s, 3H)
21	Tea	Other compounds	Lactic acid	1.30 (s, 3H)
22	Tea	Other compounds	Theanine	1.18 (t, *J* = 7.1 Hz, 3H)
23	Cannabis	Cannabinoids	Δ^8^-THC	6.08 (d, *J* = 1.7 Hz, 1H), 2.42–2.36 (m, 2H), 0.90 (t, *J* = 7.1, 1.8 Hz, 3H)
24	Cannabis	Cannabinoids	Δ^8^-THCA	6.16 (d, *J* = 1.7 Hz, 1H), 2.82–2.88 (m, 4H), 1.05 (s, 3H)
25	Cannabis	Cannabinoids	Δ^9^-THC	6.43 (t, *J* = 1.7 Hz, 1H), 1.64 (s, 3H), 1.03 (s, 3H)
26	Cannabis	Cannabinoids	Δ^9^-THCA	6.40 (t, *J* = 1.7 Hz, 1H), 1.39 (s, 3H)
27	Cannabis	Cannabinoids	CBD	6.20 (s, 1H), 4.50 (s, 1H)
28	Cannabis	Cannabinoids	CBDA	6.15 (s, 1H), 4.48 (s, 1H)
29	Cannabis	Cannabinoids	CBDVA	4.42 (s, 1H)
30	Cannabis	Cannabinoids	CBG	5.24 (t, *J* = 2.7 Hz, 1H)
31	Cannabis	Cannabinoids	CBGV	5.20 (t, *J* = 1.3 Hz, 1H)
32	Cannabis	Cannabinoids	CBN	6.14 (s, 1H)
33	Cannabis	Cannabinoids	CBC	6.62 (s, 1H)
34	Cannabis	Cannabinoids	CBCA	6.67 (s, 1H)
35	Cannabis	Carbohydrates	Sucrose	5.44 (d, *J* = 10.0 Hz, 1H)
36	Cannabis	Carbohydrates	α-Glucose	5.22 (d, *J* = 1.3 Hz, 1H)
37	Cannabis	Carbohydrates	Fructose	4.05 (d, *J* = 2.9 Hz, 1H)
38	Cannabis	Other compounds	Trigonelline	9.21 (s, 1H), 8.91 (d, *J* = 8.1 Hz, 1H), 8.84 (d, *J* = 6.1 Hz, 1H)
39	Cannabis	Other compounds	ATP	8.36 (s, 1H), 8.46 (s, 1H)
40	Cannabis	Other compounds	Choline	3.22 (s, 2H)
41	Camu Camu	Organic acids	Citric acid	2.78 (d, *J* = 15.7 Hz, 1H), 2.90 (d, *J* = 15.6 Hz, 1H)
42	Camu Camu	Organic acids	Malic acid	2.65 (dd, *J* = 16.1 Hz, 1H), 2.80 (dd, *J* = 16.1 Hz, 1H)
43	Camu Camu	Organic acids	Pyruvic acid	2.16 (s, 3H)
44	Camu Camu	Organic acids	Acetic acid	2.0 (s)
45	Camu Camu	Organic acids	Lactic acid	1.29 (s)
46	Camu Camu	Organic acids	Tartaric acid	4.30 (d, *J* = 2.8 Hz, 1H)
47	Camu Camu	Organic acids	Fumaric acid	6.61 (s)
48	Camu Camu	Organic acids	Succinic acid	2.54 (s)
49	Camu Camu	Flavanols	Flavanols derivatives	6.83 (d, *J* = 3.2 Hz, 1H), 7.57 (d, *J* = 7.8 Hz, 1H)
50	Camu Camu	Anthocyanins	Cyanidin derivatives	9.27 (s), 8.11 (s), 6.90 (s)
51	Camu Camu	Ellagic derivatives	Ellagic acid	7.36 (s, 1H)
52	Camu Camu	Gallic derivatives	Gallic acid	7.04 (s, 1H)
53	Camu Camu	Carbohydrates	α-Glucose	5.12 (d, *J* = 3.7 Hz, 1H)
54	Camu Camu	Carbohydrates	β-Glucose	4.49 (d, *J* = 7.8 Hz, 1H)
55	Camu Camu	Carbohydrates	Fructose	4.08 (d, *J* = 7.6 Hz, 1H)
56	Camu Camu	Carbohydrates	Sucrose	5.63 (d, *J* = 4.8 Hz, 1H)
57	Camu Camu	Amino acids	Alanine	1.49 (d, *J* = 2.8 Hz, 1H)
58	Camu Camu	Amino acids	Threonine	1.16 (d, *J* = 2.9 Hz, 1H)
59	Camu Camu	Amino acids	Valine	1.01 (m, 3H)
60	Camu Camu	Amino acids	Isoleucine	0.95 (m, 3H)
61	Camu Camu	Amino acids	Leucine	0.90 (m, 3H)
62	Camu Camu	Amino acids	Methionine	2.57 (t, *J* = 4.1 Hz, 1H)
63	Camu Camu	Amino acids	Glutamine	2.32–2.38 (m), 2.02–2.05 (m)
64	Camu Camu	Fatty acids	Unsaturated fatty acid	5.33 (m)
65	Camu Camu	Fatty acids	Triglycerides	4.47 (dd, *J* = 12.1, 4.0 Hz, 2H)
66	Camu Camu	Other compounds	Choline	3.20 (s)
67	Camu Camu	Other compounds	Glutamate	2.18–2.22 (m), 1.89–1.97 (m)
68	Camu Camu	Other compounds	γ-Aminobutyric acid	2.28 (t, *J* = 7.4 Hz, 1H), 2.45 (t, *J* = 7.1 Hz, 1H)

Note: Chemical shifts are reported in δ (ppm) relative to standard references (TMS). Multiplicities: s = singlet, d = doublet, t = triplet, q = quartet, m = multiplet, dd = doublet of doublets. Coupling constants (*J*) are in Hz. Data represent characteristic signals for metabolite identification in plant extracts.

## Data Availability

The original contributions presented in this study are included in the article/Appendix A. Further inquiries can be directed to the corresponding author.

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
