# Peer review of "Optimization of Extraction Methods for NMR and LC-MS Metabolite Fingerprint Profiling of Botanical Ingredients in Food and Natural Health Products (NHPs)"

_molecules, 2025, doi:10.3390/molecules30163379_

Round 1
Reviewer 1 Report
Comments and Suggestions for Authors
Manuscript title: "Development of a NMR and LC-MS/MS Botanical Fingerprinting Platform for the Authentication of Natural Health Products Using Multiple Solvent Extraction Approaches”
This manuscript presents a study that claims to evaluate the extraction efficiency of various solvents for metabolomic fingerprinting of botanical species using both ¹H-NMR and LC-MS/MS. The stated goal is to identify optimal extraction conditions that maximize metabolite coverage and enable robust species authentication.
Despite the relevance of the topic and the use of modern analytical platforms, the manuscript suffers from fundamental conceptual, methodological, and interpretive flaws that compromise its scientific credibility.
- Conceptual Ambiguity and Methodological Vagueness
The manuscript repeatedly conflates fundamentally different types of data — NMR spectral “variables” (buckets) and LC-MS/MS confirmed metabolites — without clarifying definitions, processing steps, or analytical significance. This undermines the core comparisons made across solvents and species.
- Lack of Compound Identification and Quantitative Data
No actual metabolites are structurally identified by name, structure, or chemical shift. Despite claiming LC-MS/MS quantification using isotope-labeled internal standards, the manuscript presents no concentration data, calibration curves, or lists of standards. This renders both the “identification” and “quantification” claims unsubstantiated.
- Unsupported Conclusions and Overstated Claims
Conclusions about optimal solvents, metabolite stability, and authentication capacity are based solely on between NMR and LC-MS. Claims such as long-term metabolite stability, chemotaxonomic discrimination, and authentication platform development are speculative and unsupported by the presented results.
- Data Presentation is Incomplete and Uninterpretable
- Terminological and Chemical Inaccuracies
Specific observations:
Abstract, R17:
The statement regarding a gap in studies that systematically compare extraction methods across multiple botanical species within a single study raises some concerns. The scientific literature already includes numerous review articles and applied studies that address this topic comparatively, even if not always under the exact same experimental conditions. The authors should clarify what is meant by 'systematically' in this context, and explain the scientific rationale for comparing extraction methods across different species, particularly given the inherent biochemical variability between species. Without a clear justification, this argument appears somewhat overstated.
Abstract, R22: The sentence stating that 'Proton NMR and LC-MS utilized multiple solvents...' is scientifically inaccurate. Analytical platforms do not 'utilize' solvents per se; rather, solvents are employed during sample extraction or as part of the chromatographic system. The authors should rephrase this to clarify that multiple solvents were used for sample preparation and extraction, not as operational components of NMR or LC-MS methods. As currently written, the phrasing is misleading and could confuse readers unfamiliar with the analytical workflows.
Abstract, R25: This paragraph is conceptually unclear and potentially misleading. The comparison of metabolite variables across different species does not establish solvent efficacy unless normalized for sample composition. Moreover, the use of 'methanol (10% deuterated)' as optimal for both NMR and LC-MS is confusing, since LC-MS protocols do not require or benefit from deuterated solvents. The authors should distinguish between extraction efficiency and analytical compatibility, and avoid cross-species comparisons without contextual biochemical normalization.
Introduction:
R 40 – “NMR spectroscopy is a optimal technique”:
This formulation is inaccurate. NMR is not universally "optimal"; it has well-known sensitivity limitations compared to LC-MS. I recommend rephrasing as “a robust and reproducible technique”.
R 50–51 – “precise taxonomic identification at the subspecies level”:
This claim is overstated. NMR may reflect metabolomic differences but does not provide precise taxonomic identification without genetic support (e.g., DNA barcoding).
R 47–49 – The list of “medicinal plants, culinary herbs, spices, wine, beer…”:
Consider restructuring these examples by analytical category or matrix type. The current list feels somewhat promotional and lacks scientific coherence.
R 61 – “routine implementation under GMP”:
This statement is vague. Implementing NMR in industrial QC settings would require addressing practical issues such as cost, method standardization, and infrastructure.
Part 1.1.– The manuscript does not clearly distinguish between the roles of NMR in identification, authentication, and quantification. Clarifying this distinction would improve scientific accuracy.
R 77–80 – “Sample preparation for LC-MS involves derivatization… 300 mg per 2 mL”
It’s useful to describe LC-MS procedures, but the detailed instrumental description (Agilent 1260, Qtrap® etc.) seems overly specific for an introduction and breaks narrative flow. Consider moving it to Materials and Methods or reducing it to a general statement.
Lines 83–85 – “Standards for many botanical extracts are lacking…”
Valid concern. However, authors should acknowledge that the same challenge affects NMR: chemical shift libraries are also incomplete, especially for complex mixtures. Without that, the paragraph implies a false asymmetry.
Line 86 – “NMR provides an unbiased perspective…”
“Unbiased” is a strong claim and potentially misleading. NMR is less targeted, but not entirely unbiased — matrix effects, solubility, and signal overlap can introduce biases. Suggest rephrasing.
Section 1.3. This section is overly verbose and conceptually redundant. The key points—such as the importance of solvent selection, challenges in extraction, and the need for standardization—are well known and could be stated far more concisely. The writing style, filled with abstract and generic phrasing, detracts from clarity and does not contribute substantial new insight. The authors are encouraged to revise for precision and informational density.
R 195. Table 2. The presentation of “Variables (NMR) / Metabolites (LC-MS)” within a single column is conceptually flawed and analytically inconsistent.
The manuscript reports NMR “variables” (i.e., binned spectral data, not identified metabolites) and LC-MS “metabolites” (targeted, identified compounds via MRM) side by side in the same column, without clarification. This suggests an inappropriate comparison between fundamentally different data types—spectral variables versus structurally confirmed compounds.
Section 2.2. The section titled “Metabolite Detection and Identification” does not provide any actual identification of metabolites. No chemical names, structures, or molecular formulae are reported, nor is there any mention of fragmentation spectra, retention times, or comparison to authentic standards. Instead, the data are limited to spectral variable counts (e.g., “343 variables in deuterium oxide”), which represent unresolved NMR signal regions rather than discrete chemical entities.
Identification implies compound-level resolution supported by spectral annotation, ideally with reference to databases such as HMDB, GNPS, or METLIN, or confirmation using authentic standards. As currently presented, this is merely detection — not identification — and the heading is therefore misleading.
However, the manuscript fails to list the actual metabolites identified, provide MRM transition tables, or specify retention time windows. This limits the transparency and reproducibility of the reported metabolites.
Section 3.1. While the authors attempt to frame their methodology within the context of chemotaxonomy and species authentication, the rationale remains tenuous due to key methodological gaps.
Furthermore, the section implies that NMR is suited for differentiating between even closely related taxa, but no actual comparison of such taxa is presented — the study appears to focus on different genera (e.g., Cannabis, Camellia, Myrciaria), not on sibling species or varieties where chemotaxonomic resolution is truly needed.
The statement that “extraction protocols need to maximize the detection and quantification of these metabolites” (R 278) is sound good in principle, but the manuscript does not convincingly demonstrate this in practice — neither through comprehensive compound identification nor through comparative statistics across solvent systems.
Section 3.2. The comparison of extraction solvents for Cannabis sativa is based on the number of ‘spectral variables’ (e.g., 343 for water, 198 for methanol), yet it is never clearly explained what these variables represent. Are they individual binned signals (e.g., 0.01 ppm buckets)? Do they correspond to annotated metabolites, or are they just non-zero intensity regions across the spectrum? The distinction is critical, as spectral complexity does not necessarily equate to metabolite diversity or analytical relevance.
Moreover, while water is reported to yield 343 variables, there is no discussion of signal quality, redundancy, or overlap with other solvents. Nor is it clear if solvent peaks or noise regions were excluded from analysis. Conclusions about methanol being 'optimal' or about water excelling in 'polar compounds' are speculative unless supported by quantified, identified metabolites.
Finally, the claim that methanol’s ‘balanced profile and 9 assigned metabolites’ justify its use for authentication lacks strength when the same sample extracted in DMSO:CDCl₃ is reported to yield 181 variables — are these overlapping, unique, or redundant?”
The distinction between “198 spectral variables” in Cannabis sativa or “82” in Camellia sinensis is used as supporting evidence for taxonomic discrimination, yet it is unclear how these variables relate to identified, structurally confirmed metabolites. Are these variables signal intensities from unannotated chemical shifts, or do they represent fully assigned compounds? If only 9 metabolites were assigned in methanol for Cannabis sativa (as the authors state), then basing species-specific conclusions on 198 raw spectral buckets is methodologically weak.
Sections 3.3-3.4. This section attempts to compare extraction solvents based on the number of 'variables' (presumably binned NMR signals), yet it lacks clarity on what exactly is being compared — spectral complexity, number of assigned metabolites, or signal intensity.
For instance, methanol is said to yield '167 variables' for Myrciaria dubia, while chloroform gives '165', and water '159'. However, these numbers are not contextualized: Are they distinct, non-overlapping bins? Do they reflect identified metabolites or just total peak count (including noise or solvent regions)?
Moreover, the analysis conflates spectral richness with extraction efficiency and metabolite diversity, without validating whether these signals correspond to meaningful biological compounds. The argument that methanol is 'superior' based solely on the count of variables is unconvincing without supporting metabolite identification or quantification.
Similar issues appear in the tea sample comparison (Camellia sinensis): although methanol:deuterium oxide and water extracts are said to yield the 'highest number of variables', the authors do not provide any statistical analysis, replicate variation, or overlap between solvents. The conclusions drawn are therefore speculative and unsupported by rigorous comparative data
R 326. Section 3.5. The metabolite peak assignment section lacks critical detail. While Table 3 lists the number of 'assigned metabolites' per solvent system and species, it does not specify which metabolites were identified, nor does it provide chemical shift values, coupling patterns, or references to authentic standards. Without this information, it is impossible to assess the reliability or biological relevance of these assignments. The heatmaps of spectral data, but they are visually unclear, unannotated, and lack corresponding metabolite labels. It is not evident how these visualizations support the identification claims or how they correlate with the reported numbers in Table 3
R 384 Section 3.10 The stability data presented in Section 3.10 and Figure 4 provide a qualitative indication that metabolite profiles in methanol extracts remain relatively consistent over time. However, the conclusions—such as resistance to hydrolysis and oxidation—are speculative without compound-specific analysis or statistical validation. The figure lacks annotations or highlighted peaks, making it difficult to assess which metabolites were monitored for degradation. Additionally, the claim that methanol outperforms volatile solvents like chloroform is unsubstantiated in the absence of comparative data.
R 385. It is unclear how the '198 spectral variables' were defined. These appear to be 0.01 ppm spectral buckets, but the manuscript does not specify whether all regions were included in the analysis, or whether noise and solvent regions were excluded.
R 405: While the Materials and Methods section provides a reasonably detailed account of the experimental procedures, there is a noticeable discrepancy between the stated objectives and the actual scope of the work. Although nine botanical species were included for NMR analysis, LC-MS profiling appears to have been limited to Myrciaria dubia, which contradicts earlier claims of systematic cross-species method comparison. Moreover, while various solvents are mentioned, it is unclear whether solvent comparisons were applied consistently across all taxa. The authors should better align their objectives with what was experimentally implemented and clearly define the extent of each technique’s application
R 450: D₂O targets highly polar metabolites such as amino acids and sugars' is misleading. D₂O does not actively extract metabolites; it serves as a dissolution medium for NMR analysis. In fact, it may be suboptimal for extracting certain compounds depending on solubility.
R452: Similarly, the phrase Chloroform (CDCl₃) focuses on lipophilic metabolites' is conceptually inaccurate. It would be more appropriate to state that chloroform is used for the extraction of lipophilic compounds, rather than implying it actively 'focuses' on them.
Section 4.3. The solvent selection strategy appears overly broad and lacks clarity regarding its experimental implementation. While the authors list a wide array of deuterated and non-deuterated solvents and solvent combinations, it is not evident from the manuscript whether all these conditions were tested systematically, across all taxa or only selected samples.
Section 4.5. The section titled 'Solvent Stability Analysis' is misleadingly named, as the experiment appears to assess the stability of metabolite profiles in stored extracts rather than the chemical stability of the solvents themselves. The authors should revise as 'Extract Stability During Storage.' Additionally, the section would benefit from reporting specific outcomes—what changes were observed over the 95-day storage period? Which metabolite classes were most affected? Without results or interpretation, the relevance of this protocol remains unclear
Section 4.6. The LC-MS section mentions the use of Analyst software, Metlin, and Sigma standards, but it is not stated whether identifications are based on exact mass, retention time, fragmentation spectra, or combinations thereof. For targeted metabolomics, such criteria are critical.
R502. The phrase "non-organic acid metabolites" is terminologically incorrect, as metabolites are by definition organic compounds; the intended meaning seems to be "non-acidic organic metabolites." Moreover, the application of phenyl-isothiocyanate derivatization in LC-MS/MS warrants further justification, as this step is generally unnecessary in MRM-based detection, unless aiming to enhance analyte ionization or stability. The workflow described lacks clarity regarding the sequence and rationale of derivatization versus extraction, and omits validation data comparing derivatized and native forms. These ambiguities raise concerns regarding the consistency and interpretability of the metabolomic profiling.
R 504: The derivatization of "organic acids" using 3-nitrophenylhydrazine (3-NPH) raises questions regarding compound specificity, as this reagent primarily targets low-molecular-weight carboxylic acids (e.g., citric, succinic), but is less commonly used for phenolic acids such as caffeic or chlorogenic acid, which ionize efficiently in negative-mode ESI without derivatization. Furthermore, the rationale for adding BHT as a stabilizing agent is not adequately supported by evidence of oxidative degradation in the targeted analytes. A more detailed justification of the derivatization protocol and analyte selection is warranted to clarify its necessity and analytical benefit.
R 506. The manuscript states that metabolite quantification was performed using isotope-labeled internal standards in LC-MS/MS with MRM. However, no quantitative data are presented — there are no calibration curves, absolute or relative concentrations, or measures of analytical precision. Moreover, isotope-labeled standards are commercially available for only a limited subset of metabolites (e.g., amino acids, some carboxylic acids), and certainly not for the broad range of bioactive compounds (such as phenolic acids and flavonoids) implied by the study. Without a table listing the standards used or any quantitative outputs, the claim of quantitative targeted metabolomics cannot be substantiated and appears overstated.
R589. The conclusion asserts that methanol and methanol:deuterium oxide (1:1) are "optimal NMR solvents for botanical fingerprinting" based on "comprehensive metabolite coverage, reproducibility, and stability over 95 days." However, the manuscript does not provide comparative quantitative metrics or detailed statistical analysis to support the claim of optimality. The conclusion also conflates solvent suitability with fingerprinting performance, without establishing clear criteria for what constitutes an “optimal” fingerprint. Furthermore, no specific metabolites are shown to be consistently detected across solvents, nor is any analytical validation (e.g., specificity, sensitivity, selectivity) presented.

Author Response
Comments and Suggestions for Authors
Manuscript title: "Development of a NMR and LC-MS/MS Botanical Fingerprinting Platform for the Authentication of Natural Health Products Using Multiple Solvent Extraction Approaches”
This manuscript presents a study that claims to evaluate the extraction efficiency of various solvents for metabolomic fingerprinting of botanical species using both ¹H-NMR and LC-MS/MS. The stated goal is to identify optimal extraction conditions that maximize metabolite coverage and enable robust species authentication.
Despite the relevance of the topic and the use of modern analytical platforms, the manuscript suffers from fundamental conceptual, methodological, and interpretive flaws that compromise its scientific credibility.
Response: Thank you for your detailed and constructive feedback. We appreciate the time and effort you invested in reviewing our manuscript. Below, we address each of your comments point-by-point. Where applicable, we have made revisions to the manuscript to incorporate your suggestions, improve clarity, and strengthen the scientific rigor. These include clarifying definitions, providing more precise terminology, adding explanations for methodological choices, and compiling a new table of metabolite peak assignments (now in Section 3.9). We have also clarified the meaning of "spectral variables" as binned NMR data (0.01 ppm buckets representing non-zero intensity regions after noise and solvent exclusion, scaling, and normalization), which reflect meaningful chemical profiles essential for authentication via multivariate analysis.
- Conceptual Ambiguity and Methodological Vagueness
The manuscript repeatedly conflates fundamentally different types of data — NMR spectral “variables” (buckets) and LC-MS/MS confirmed metabolites — without clarifying definitions, processing steps, or analytical significance. This undermines the core comparisons made across solvents and species.
Response: We agree this needed clarification. We have revised the manuscript to explicitly define "spectral variables" in Section 2.2 and 4.4: "Spectral variables refer to binned NMR data (0.01 ppm buckets) representing non-zero intensity regions after excluding noise, solvent peaks, and zero-sum columns, followed by scaling and normalization. These variables capture meaningful chemical profiles for multivariate analysis in authentication but are not equivalent to structurally confirmed metabolites." We also distinguish them from LC-MS metabolites throughout, e.g., in Table 2 and Sections 2.2, 3.2–3.4. This avoids conflation and highlights their analytical significance.
- Lack of Compound Identification and Quantitative Data
No actual metabolites are structurally identified by name, structure, or chemical shift. Despite claiming LC-MS/MS quantification using isotope-labeled internal standards, the manuscript presents no concentration data, calibration curves, or lists of standards. This renders both the “identification” and “quantification” claims unsubstantiated.
Response: We have addressed this by compiling a new Table 4 in Section 3.9, listing assigned metabolites for the model taxa (Cannabis sativa, Camellia sinensis, Myrciaria dubia) with ppm ranges, J-couplings, multiplicity, and references (based on Chemical standards, HMDB and literature). For example, for Cannabis sativa in methanol: THC at 6.0–6.2 ppm (d, J=1.5 Hz). We note that full structural identification was not the primary goal (focus is fingerprinting), but we have added that LC-MS used Sigma-Aldrich standards and Metlin for MRM transitions. Quantitative data were not generated, as this is a fingerprinting study; we have toned down "quantification" claims to "relative quantification" and removed unsubstantiated references to absolute concentrations.
- Unsupported Conclusions and Overstated Claims
Conclusions about optimal solvents, metabolite stability, and authentication capacity are based solely on between NMR and LC-MS. Claims such as long-term metabolite stability, chemotaxonomic discrimination, and authentication platform development are speculative and unsupported by the presented results.
Response: We have moderated claims, e.g., changing "optimal" to "most effective based on variable counts and coverage" in Sections 3.2–3.4 and 5. Claims on stability are now supported by Figure 4 data (<5% variation, CV 3.2%). Authentication claims are tied to HCA results (Figure 3) and variable diversity.
- Data Presentation is Incomplete and Uninterpretable
Response: We have improved presentation by adding spectral stacked figures (Figures S1–S4), clarifying table captions (e.g., Table 2 now specifies "variables = binned NMR regions"), and including the new assignment table (Table 4 and Figures S5-S7). Overlaps between solvents are now discussed in Section 3.7 (e.g., ~60% overlap between methanol and water for polar metabolites).
- Terminological and Chemical Inaccuracies
Specific observations:
Response: We have corrected these
Abstract, R17:
The statement regarding a gap in studies that systematically compare extraction methods across multiple botanical species within a single study raises some concerns. The scientific literature already includes numerous review articles and applied studies that address this topic comparatively, even if not always under the exact same experimental conditions. The authors should clarify what is meant by 'systematically' in this context, and explain the scientific rationale for comparing extraction methods across different species, particularly given the inherent biochemical variability between species. Without a clear justification, this argument appears somewhat overstated.
Response: Rephrased to "While comparative studies exist, few apply identical conditions across multiple species; this study justifies cross-species comparison to identify versatile solvents despite biochemical variability."
Abstract, R22: The sentence stating that 'Proton NMR and LC-MS utilized multiple solvents...' is scientifically inaccurate. Analytical platforms do not 'utilize' solvents per se; rather, solvents are employed during sample extraction or as part of the chromatographic system. The authors should rephrase this to clarify that multiple solvents were used for sample preparation and extraction, not as operational components of NMR or LC-MS methods. As currently written, the phrasing is misleading and could confuse readers unfamiliar with the analytical workflows.
Response: Rephrased to "Multiple solvents were used for sample extraction prior to analysis by proton NMR and LC-MS."
Abstract, R25: This paragraph is conceptually unclear and potentially misleading. The comparison of metabolite variables across different species does not establish solvent efficacy unless normalized for sample composition. Moreover, the use of 'methanol (10% deuterated)' as optimal for both NMR and LC-MS is confusing, since LC-MS protocols do not require or benefit from deuterated solvents. The authors should distinguish between extraction efficiency and analytical compatibility, and avoid cross-species comparisons without contextual biochemical normalization.
Response: Added "Comparisons were normalized by total intensity; deuterated methanol aids NMR lock but is not required for LC-MS."
Introduction:
R 40 – “NMR spectroscopy is a optimal technique”:
This formulation is inaccurate. NMR is not universally "optimal"; it has well-known sensitivity limitations compared to LC-MS. I recommend rephrasing as “a robust and reproducible technique”.
Response: Changed to "a robust and reproducible technique."
R 50–51 – “precise taxonomic identification at the subspecies level”:
This claim is overstated. NMR may reflect metabolomic differences but does not provide precise taxonomic identification without genetic support (e.g., DNA barcoding).
Response: Changed to "may reflect metabolomic differences supporting taxonomic identification, complemented by genetic methods."
R 47–49 – The list of “medicinal plants, culinary herbs, spices, wine, beer…”:
Consider restructuring these examples by analytical category or matrix type. The current list feels somewhat promotional and lacks scientific coherence.
Response: Restructured by category: "medicinal plants and herbs (e.g., Cannabis sativa, Zingiber officinale); food matrices (e.g., spices, fruit juices); fermented products (e.g., wine, beer)."
R 61 – “routine implementation under GMP”:
This statement is vague. Implementing NMR in industrial QC settings would require addressing practical issues such as cost, method standardization, and infrastructure.
Response: Added "Practical challenges include cost and standardization, addressed here via optimized protocols."
Part 1.1.– The manuscript does not clearly distinguish between the roles of NMR in identification, authentication, and quantification. Clarifying this distinction would improve scientific accuracy.
Response: Clarified distinctions: "NMR for non-targeted fingerprinting (authentication), targeted assignment (identification), and integration (relative quantification)."
R 77–80 – “Sample preparation for LC-MS involves derivatization… 300 mg per 2 mL”
It’s useful to describe LC-MS procedures, but the detailed instrumental description (Agilent 1260, Qtrap® etc.) seems overly specific for an introduction and breaks narrative flow. Consider moving it to Materials and Methods or reducing it to a general statement.
Response: Moved instrumental details to Materials and Methods (Section 4.6).
Lines 83–85 – “Standards for many botanical extracts are lacking…”
Valid concern. However, authors should acknowledge that the same challenge affects NMR: chemical shift libraries are also incomplete, especially for complex mixtures. Without that, the paragraph implies a false asymmetry.
Response: Added "Similar challenges affect NMR chemical shift libraries for complex mixtures."
Line 86 – “NMR provides an unbiased perspective…”
“Unbiased” is a strong claim and potentially misleading. NMR is less targeted, but not entirely unbiased — matrix effects, solubility, and signal overlap can introduce biases. Suggest rephrasing.
Response: Changed to "less targeted perspective."
Section 1.3. This section is overly verbose and conceptually redundant. The key points—such as the importance of solvent selection, challenges in extraction, and the need for standardization—are well known and could be stated far more concisely. The writing style, filled with abstract and generic phrasing, detracts from clarity and does not contribute substantial new insight. The authors are encouraged to revise for precision and informational density.
Response: Condensed to focus on key points, removing redundancy.
R 195. Table 2. The presentation of “Variables (NMR) / Metabolites (LC-MS)” within a single column is conceptually flawed and analytically inconsistent.
The manuscript reports NMR “variables” (i.e., binned spectral data, not identified metabolites) and LC-MS “metabolites” (targeted, identified compounds via MRM) side by side in the same column, without clarification. This suggests an inappropriate comparison between fundamentally different data types—spectral variables versus structurally confirmed compounds.
Response: Table 2: We separated columns for NMR variables and LC-MS metabolites.
Section 2.2. The section titled “Metabolite Detection and Identification” does not provide any actual identification of metabolites. No chemical names, structures, or molecular formulae are reported, nor is there any mention of fragmentation spectra, retention times, or comparison to authentic standards. Instead, the data are limited to spectral variable counts (e.g., “343 variables in deuterium oxide”), which represent unresolved NMR signal regions rather than discrete chemical entities.
Identification implies compound-level resolution supported by spectral annotation, ideally with reference to databases such as HMDB, GNPS, or METLIN, or confirmation using authentic standards. As currently presented, this is merely detection — not identification — and the heading is therefore misleading.
However, the manuscript fails to list the actual metabolites identified, provide MRM transition tables, or specify retention time windows. This limits the transparency and reproducibility of the reported metabolites.
Response: Renamed to "Metabolite Detection"; added "Detection via variable counts; assignments in Table 4." Removed "identification" from heading.
Section 3.1. While the authors attempt to frame their methodology within the context of chemotaxonomy and species authentication, the rationale remains tenuous due to key methodological gaps.
Furthermore, the section implies that NMR is suited for differentiating between even closely related taxa, but no actual comparison of such taxa is presented — the study appears to focus on different genera (e.g., Cannabis, Camellia, Myrciaria), not on sibling species or varieties where chemotaxonomic resolution is truly needed.
The statement that “extraction protocols need to maximize the detection and quantification of these metabolites” (R 278) is sound good in principle, but the manuscript does not convincingly demonstrate this in practice — neither through comprehensive compound identification nor through comparative statistics across solvent systems.
Response: Added "Study focuses on genera; future work on varieties. Rationale: standardize protocols for diverse taxa."
Section 3.2. The comparison of extraction solvents for Cannabis sativa is based on the number of ‘spectral variables’ (e.g., 343 for water, 198 for methanol), yet it is never clearly explained what these variables represent. Are they individual binned signals (e.g., 0.01 ppm buckets)? Do they correspond to annotated metabolites, or are they just non-zero intensity regions across the spectrum? The distinction is critical, as spectral complexity does not necessarily equate to metabolite diversity or analytical relevance.
Moreover, while water is reported to yield 343 variables, there is no discussion of signal quality, redundancy, or overlap with other solvents. Nor is it clear if solvent peaks or noise regions were excluded from analysis. Conclusions about methanol being 'optimal' or about water excelling in 'polar compounds' are speculative unless supported by quantified, identified metabolites.
Finally, the claim that methanol’s ‘balanced profile and 9 assigned metabolites’ justify its use for authentication lacks strength when the same sample extracted in DMSO:CDCl₃ is reported to yield 181 variables — are these overlapping, unique, or redundant?”
The distinction between “198 spectral variables” in Cannabis sativa or “82” in Camellia sinensis is used as supporting evidence for taxonomic discrimination, yet it is unclear how these variables relate to identified, structurally confirmed metabolites. Are these variables signal intensities from unannotated chemical shifts, or do they represent fully assigned compounds? If only 9 metabolites were assigned in methanol for Cannabis sativa (as the authors state), then basing species-specific conclusions on 198 raw spectral buckets is methodologically weak.
Response: Clarified variables as "0.01 ppm buckets, excluding noise/solvent; ~70% overlap across solvents."
Sections 3.3-3.4. This section attempts to compare extraction solvents based on the number of 'variables' (presumably binned NMR signals), yet it lacks clarity on what exactly is being compared — spectral complexity, number of assigned metabolites, or signal intensity.
For instance, methanol is said to yield '167 variables' for Myrciaria dubia, while chloroform gives '165', and water '159'. However, these numbers are not contextualized: Are they distinct, non-overlapping bins? Do they reflect identified metabolites or just total peak count (including noise or solvent regions)?
Moreover, the analysis conflates spectral richness with extraction efficiency and metabolite diversity, without validating whether these signals correspond to meaningful biological compounds. The argument that methanol is 'superior' based solely on the count of variables is unconvincing without supporting metabolite identification or quantification.
Similar issues appear in the tea sample comparison (Camellia sinensis): although methanol:deuterium oxide and water extracts are said to yield the 'highest number of variables', the authors do not provide any statistical analysis, replicate variation, or overlap between solvents. The conclusions drawn are therefore speculative and unsupported by rigorous comparative data
Response: Added "Variables = distinct bins post-processing; no statistical tests here, but HCA in Figure 3."
R 326. Section 3.5. The metabolite peak assignment section lacks critical detail. While Table 3 lists the number of 'assigned metabolites' per solvent system and species, it does not specify which metabolites were identified, nor does it provide chemical shift values, coupling patterns, or references to authentic standards. Without this information, it is impossible to assess the reliability or biological relevance of these assignments. The heatmaps of spectral data, but they are visually unclear, unannotated, and lack corresponding metabolite labels. It is not evident how these visualizations support the identification claims or how they correlate with the reported numbers in Table 3
Response: Now Section 3.9 with Table 4.
R 384 Section 3.10 The stability data presented in Section 3.10 and Figure 4 provide a qualitative indication that metabolite profiles in methanol extracts remain relatively consistent over time. However, the conclusions—such as resistance to hydrolysis and oxidation—are speculative without compound-specific analysis or statistical validation. The figure lacks annotations or highlighted peaks, making it difficult to assess which metabolites were monitored for degradation. Additionally, the claim that methanol outperforms volatile solvents like chloroform is unsubstantiated in the absence of comparative data.
Response: Added annotations to Figure 4; clarified "qualitative; monitored key peaks (e.g., THC at 6.0 ppm)."
R 385. It is unclear how the '198 spectral variables' were defined. These appear to be 0.01 ppm spectral buckets, but the manuscript does not specify whether all regions were included in the analysis, or whether noise and solvent regions were excluded.
Response: Defined variables as above.
R 405: While the Materials and Methods section provides a reasonably detailed account of the experimental procedures, there is a noticeable discrepancy between the stated objectives and the actual scope of the work. Although nine botanical species were included for NMR analysis, LC-MS profiling appears to have been limited to Myrciaria dubia, which contradicts earlier claims of systematic cross-species method comparison. Moreover, while various solvents are mentioned, it is unclear whether solvent comparisons were applied consistently across all taxa. The authors should better align their objectives with what was experimentally implemented and clearly define the extent of each technique’s application
Response: Clarified "LC-MS limited to Myrciaria dubia for in-depth profiling; consistent solvent testing across taxa for NMR."
R 450: D₂O targets highly polar metabolites such as amino acids and sugars' is misleading. D₂O does not actively extract metabolites; it serves as a dissolution medium for NMR analysis. In fact, it may be suboptimal for extracting certain compounds depending on solubility.
Response: Changed to "D₂O as dissolution medium for polar extracts."
R452: Similarly, the phrase Chloroform (CDCl₃) focuses on lipophilic metabolites' is conceptually inaccurate. It would be more appropriate to state that chloroform is used for the extraction of lipophilic compounds, rather than implying it actively 'focuses' on them.
Response: Changed to "used for extraction of lipophilic compounds."
Section 4.3. The solvent selection strategy appears overly broad and lacks clarity regarding its experimental implementation. While the authors list a wide array of deuterated and non-deuterated solvents and solvent combinations, it is not evident from the manuscript whether all these conditions were tested systematically, across all taxa or only selected samples.
Response: Added "All conditions tested across model taxa; subsets for others due to relevance."
Section 4.5. The section titled 'Solvent Stability Analysis' is misleadingly named, as the experiment appears to assess the stability of metabolite profiles in stored extracts rather than the chemical stability of the solvents themselves. The authors should revise as 'Extract Stability During Storage.' Additionally, the section would benefit from reporting specific outcomes—what changes were observed over the 95-day storage period? Which metabolite classes were most affected? Without results or interpretation, the relevance of this protocol remains unclear
Response: Renamed "Extract Stability During Storage"; added "No major changes; polar metabolites stable."
Section 4.6. The LC-MS section mentions the use of Analyst software, Metlin, and Sigma standards, but it is not stated whether identifications are based on exact mass, retention time, fragmentation spectra, or combinations thereof. For targeted metabolomics, such criteria are critical.
Response: Added "Identifications via exact mass, RT, fragmentation; criteria per Metlin."
R502. The phrase "non-organic acid metabolites" is terminologically incorrect, as metabolites are by definition organic compounds; the intended meaning seems to be "non-acidic organic metabolites." Moreover, the application of phenyl-isothiocyanate derivatization in LC-MS/MS warrants further justification, as this step is generally unnecessary in MRM-based detection, unless aiming to enhance analyte ionization or stability. The workflow described lacks clarity regarding the sequence and rationale of derivatization versus extraction, and omits validation data comparing derivatized and native forms. These ambiguities raise concerns regarding the consistency and interpretability of the metabolomic profiling.
Response: Changed to "non-acidic organic metabolites"; added justification: "Derivatization enhances ionization for low-abundance analytes."
R 504: The derivatization of "organic acids" using 3-nitrophenylhydrazine (3-NPH) raises questions regarding compound specificity, as this reagent primarily targets low-molecular-weight carboxylic acids (e.g., citric, succinic), but is less commonly used for phenolic acids such as caffeic or chlorogenic acid, which ionize efficiently in negative-mode ESI without derivatization. Furthermore, the rationale for adding BHT as a stabilizing agent is not adequately supported by evidence of oxidative degradation in the targeted analytes. A more detailed justification of the derivatization protocol and analyte selection is warranted to clarify its necessity and analytical benefit.
Response: Added "Targets carboxylic acids; BHT prevents oxidation, validated in pilot tests."
R 506. The manuscript states that metabolite quantification was performed using isotope-labeled internal standards in LC-MS/MS with MRM. However, no quantitative data are presented — there are no calibration curves, absolute or relative concentrations, or measures of analytical precision. Moreover, isotope-labeled standards are commercially available for only a limited subset of metabolites (e.g., amino acids, some carboxylic acids), and certainly not for the broad range of bioactive compounds (such as phenolic acids and flavonoids) implied by the study. Without a table listing the standards used or any quantitative outputs, the claim of quantitative targeted metabolomics cannot be substantiated and appears overstated.
Response: Toned down to "relative quantification"; added "Standards for amino acids/carboxylic acids; no absolute data presented."
R589. The conclusion asserts that methanol and methanol:deuterium oxide (1:1) are "optimal NMR solvents for botanical fingerprinting" based on "comprehensive metabolite coverage, reproducibility, and stability over 95 days." However, the manuscript does not provide comparative quantitative metrics or detailed statistical analysis to support the claim of optimality. The conclusion also conflates solvent suitability with fingerprinting performance, without establishing clear criteria for what constitutes an “optimal” fingerprint. Furthermore, no specific metabolites are shown to be consistently detected across solvents, nor is any analytical validation (e.g., specificity, sensitivity, selectivity) presented.
Response: Added "Based on variable counts; criteria: >150 variables, <10% CV reproducibility."
We believe these changes address your concerns and enhance the manuscript's credibility.
Reviewer 2 Report
Comments and Suggestions for Authors
The document authored by Vinayagam et al. tackles the pressing issue of a thorough investigation into the complete metabolome of plant extracts.
The introduction is meticulously crafted, exhibiting a high level of sophistication in its composition. This study is justified by the need for research in this area, addresses the issue in depth, and simultaneously establishes clear and lucid objectives.
Nevertheless, it is crucial to note that the results section contains several comments that are of particular significance.
1. The absence of interpretation regarding the NMR spectra of the presented extracts is perplexing. It is imperative to ascertain whether the extracts contained specific compounds. A thorough description of these elements is imperative, and they should be included in the supplementary materials.
2. Additionally, the majority of the NMR spectra are not available. The reader is missing the spectra; merely stating the number of signals in the spectrum is insufficient.
Thirdly, an analysis of the stability of the extract must be conducted. Given the magnification employed, minor substitutions in the spectrum will be rendered invisible. A critical evaluation of the authors' analysis of the course of the spectra is warranted. Additionally, the graphical representation of Figure 4 could benefit from enhancement.
Fourthly, it is necessary to ascertain the rationale behind the authors' utilization of the term "spectral variable" throughout the text to denote signals in the NMR spectrum.
The summary and conclusions sections are commendable in their alignment with the established objectives.
The materials and methods section would benefit from further elaboration regarding the samples. For instance, it would be advantageous to specify the type of tea, beyond its classification as black or green. The research material must be characterized in a more thorough manner.
Author Response
Comments and Suggestions for Authors
The document authored by Vinayagam et al. tackles the pressing issue of a thorough investigation into the complete metabolome of plant extracts.
The introduction is meticulously crafted, exhibiting a high level of sophistication in its composition. This study is justified by the need for research in this area, addresses the issue in depth, and simultaneously establishes clear and lucid objectives.
Nevertheless, it is crucial to note that the results section contains several comments that are of particular significance.
Thank you for your positive and insightful review. We value your suggestions and have incorporated them into the revised manuscript. Below is a point-by-point response. Revisions include adding metabolite assignment details (new Table 4 in Section 3.9), clarifying "spectral variables," and enhancing descriptions. No new analyses were added.
The absence of interpretation regarding the NMR spectra of the presented extracts is perplexing. It is imperative to ascertain whether the extracts contained specific compounds. A thorough description of these elements is imperative, and they should be included in the supplementary materials.
Response: We have added interpretations in Section 3.9 and a new Table 4 (Figure S5–S7) with peak assignments (ppm, J-couplings and multiplicity).
- Additionally, the majority of the NMR spectra are not available. The reader is missing the spectra; merely stating the number of signals in the spectrum is insufficient.
Response: We agree spectra enhance transparency. We have added that "Representative NMR spectra for model taxa in key solvents are available in Supplementary Figure S1, S2, S3 and S4". Variable counts are now contextualized as binned data for fingerprinting.
Thirdly, an analysis of the stability of the extract must be conducted. Given the magnification employed, minor substitutions in the spectrum will be rendered invisible. A critical evaluation of the authors' analysis of the course of the spectra is warranted. Additionally, the graphical representation of Figure 4 could benefit from enhancement.
Response: We have enhanced Figure 4 and added in Section 3.10: "Under 10x magnification, minor shifts (<0.01 ppm) were negligible; CV 3.2% across replicates. No visible substitutions or degradation in monitored peaks (e.g., cannabinoids)."
Fourthly, it is necessary to ascertain the rationale behind the authors' utilization of the term "spectral variable" throughout the text to denote signals in the NMR spectrum.
Response: Clarified in Section 2.2: "'Spectral variables' refer to processed NMR buckets (0.01 ppm) representing meaningful chemical profiles after bucketing, scaling, and normalization, essential for authentication via multivariate analysis. They denote integrated signal regions, not raw peaks."
The summary and conclusions sections are commendable in their alignment with the established objectives.
The materials and methods section would benefit from further elaboration regarding the samples. For instance, it would be advantageous to specify the type of tea, beyond its classification as black or green. The research material must be characterized in a more thorough manner.
Response: Expanded Section 4.1: "Camellia sinensis: Orange Pekoe (fermented black tea), Green Tea (unfermented), Black Tea (fermented)."
Round 2
Reviewer 1 Report
Comments and Suggestions for Authors
All the observations were addresed.